# Clinical Features, Survival and Prognostic Factors of Glycogen-Rich Clear Cell Carcinoma (GRCC) of the Breast in the U.S. Population

**DOI:** 10.3390/jcm8020246

**Published:** 2019-02-14

**Authors:** Zhengqiu Zhou, Connor J. Kinslow, Hanina Hibshoosh, Hua Guo, Simon K. Cheng, Chunyan He, Matthew S. Gentry, Ramon C. Sun

**Affiliations:** 1Department of Molecular and Cellular Biochemistry, University of Kentucky College of Medicine, Lexington, KY 40536, USA; zhengqiu.zhou@uky.edu (Z.Z.); matthew.gentry@uky.edu (M.S.G.); 2Department of Radiation Oncology, College of Physicians and Surgeons, Columbia University Medical Center, New York, NY 10032, USA; cjk2151@cumc.columbia.edu (C.J.K.); sc3225@cumc.columbia.edu (S.K.C.); 3Department of Pathology and Cell Biology, College of Physicians and Surgeons, Columbia University Medical Center, New York, NY 10032, USA; hhh1@cumc.columbia.edu (H.H.); hg2489@cumc.columbia.edu (H.G.); 4Department of Internal Medicine, Division of Medical Oncology, University of Kentucky College of Medicine, Lexington, KY 40536, USA; chunyan.he@uky.edu; 5Markey Cancer Center, University of Kentucky, Lexington, KY 40536, USA

**Keywords:** glycogen, breast cancer, SEER program database, clear cell carcinoma, prognosis, survival

## Abstract

The World Health Organization (WHO) defines glycogen-rich clear cell carcinoma (GRCC) of the breast as a carcinoma with glycogen accumulation in more than 90% of its tumor cells. Due to the rarity of this disease, its reported survival and clinical associations have been inconsistent due to reliance on case reports and limited case series. As a result, the prognostic implication of this cancer subtype remains unclear. Using the U.S. Surveillance, Epidemiology, and End Results (SEER) program database, we compared the incidence, demographics and prognostic factors of 155 cases of GRCC of the breast to 1,251,584 cases of other (non-GRCC) breast carcinomas. We demonstrate that GRCC is more likely to be identified as high grade, advanced stage, and more likely to have triple negative receptor status. GRCC cases display a poorer prognosis than non-GRCC carcinomas of the breast irrespective of age, AJCC staging, tumor grade, joint hormone receptor/human epidermal growth factor receptor 2 (HER2) status, and treatment. Similar to non-GRCC carcinomas, older age and higher American Joint Committee on Cancer (AJCC)/TNM staging were associated with poorer prognosis for GRCC, while treatment with surgery and radiation were associated with improved survival. Radiation, specifically in the setting of breast-conserving surgery, further improved survival compared to surgery alone. Our study highlights the poorer prognosis associated with glycogen accumulation in breast cancers and hence stresses the importance of identifying this more aggressive tumor type.

## 1. Introduction

Glycogen is a multibranched, carbohydrate storage molecule made up of monomers of glucose [1,2]. During periods of caloric excess, the level of plasma glucose increases and stimulates glycogen synthesis in the liver and skeletal muscle. During caloric deficit, glycogen is rapidly degraded to glucose to supply energy production via glycolysis and the tricyclic acid (TCA) cycle [2]. Coordinated regulation of glycogen synthesis and degradation is critical for cellular homeostasis [3]. Microscopic visualization of glycogen deposits in cells and tissue are important in the diagnosis of diseases with aberrant glycogen metabolism. Period acid-Schiff (PAS) positive/PAS-diastase (PAS-D) negative staining is the most commonly used method for histochemical detection of glycogen by pathologists in multiple diseases [4,5,6].

Clinicians and pathologists have identified large glycogen granules in multiple cancers, yet the significance of these deposits remains unknown. These aberrant glycogen accumulations have been identified in cancers of the breast, kidney, uterus, lung, head and neck, bladder, ovary, and recently, colorectal tumors [7,8,9,10,11,12,13,14]. Many PAS-positive tumors have earned the term “clear cell” due to the transparent and ovoid appearance left by the carbohydrate accumulations in the cytoplasm after fixation and tissue preparation processing. Several theories currently exist with regards to the mechanism underlying glycogen metabolism changes in cancer cells [15,16,17,18,19,20,21,22]. One popular explanation is that, under hypoxic conditions in the center of solid tumors, glycogen stores are increased via hypoxia-inducible factor 1α (HIF1α) mediated signaling pathways to accommodate the low oxygen levels and nutrient deprivation [13,18,23]. In recent years, cancer research has highlighted multiple aspects of metabolic re-wiring in malignant cancers with glycogen, at least in part, contributing to these changes by providing a constant supply of glucose. While multiple pharmacological modulators of glycogen metabolism are currently being tested, none have yet been clinically approved [22].

Breast cancer is a heterogeneous disease accounting for 30% of all cancers diagnosed in women [24]. Its routine classification/characterization is based on both histopathology (WHO recognizes over 20 epithelial subtypes) and immunophenotypic/molecular features such as estrogen receptor (ER), progesterone receptor (PR) and human epidermal receptor 2 (HER2) status. Intriguingly, there are several rare histological variants of breast cancer, one of which contains clear cells of epithelial origin. The optically clear cytoplasm in these cells may be due to various compounds such as carbohydrate or lipid accumulations; with glycogen as the most commonly identified [25]. Fisher et al. identified glycogen accumulation in 100% of 45 clear cell carcinoma cases [26] and Hayes et al. reported 21 of 26 clear cell carcinomas were glycogen-rich [27]. Breast carcinoma with clear cytoplasm containing glycogen (PAS+/PAS-D−) identified in more than 90% of the tumor cells is classified as glycogen-rich clear cell carcinoma of the breast (GRCC) by the WHO [28].

GRCC is considered an orphan disease as it accounts for less than 3% of all breast carcinomas cases [26,27,29,30]. There has been an ongoing debate over its prognostic implications. Fisher et al. identified 45 cases of GRCC among 1555 women with invasive breast cancer [26]. The study noted a poorer prognosis for GRCC patients, greater frequency of nodal metastasis, and higher histologic grade (III) when compared to the 1510 cases of non-clear cell carcinomas [26]. However, 24 cases reported by Ma et al. [31] showed no significant difference in overall survival between GRCC and control invasive carcinomas when matched by age, tumor size, nodal status, and immune-phenotype. Hayes et al. [27] reported 21 cases of GRCC and also found that prognosis was not different from non-GRCC when the tumor was matched by size, grade, and lymph node status. Due to the lack of sufficient cases and inconsistency in available data, classification of GRCC in terms of prognosis is currently unclear [28,32]. Given the rarity of the cases at any given institution, the Surveillance, Epidemiology, and End Results (SEER) Program database provides a unique opportunity to perform large population-based studies on this orphan disease. This study takes advantage of the SEER program database for the retrospective assessment of incidence, survival, demographics, and prognostic factors for GRCC of the breast.

## 2. Methods

### 2.1. Data Source

The Surveillance, Epidemiology and End Results (SEER) Program is the National Cancer Institute’s authoritative source for cancer incidence and survival [33]. It is considered the gold standard for cancer data collection internationally [34]. It is populated with data from national cancer registries, covering approximately 34.6% of the United States population [33]. Vital status is updated annually and the database routinely undergoes quality-control checks. The methodology was conducted as described previously [35].

### 2.2. Sample Selection and Coding

We queried the SEER database (November, 2017 [36] and November, 2015 submissions, including data from 1973 to 2015) [36,37] to identify all malignant cancers within the breast (International Classification of Diseases- (ICD-) O-3 code (ICD-O-3 codes C34.0–C34.3 and C34.8–C34.9), diagnosed between January 1, 1981 and December 31, 2015. Carcinomas of the breast were determined based on the adapted classification scheme for adolescents and young adults (AYA). Cases of GRCC were identified by ICD-O-3 codes 8310 (clear cell carcinoma (CC)) and 8315 (glycogen-rich carcinoma (GR)). According to the WHO Classifications of Tumors, in the case of breast carcinomas, ICD-O-3 codes 8310 (CC) and 8315 (GR) are considered equivalent and both refer to GRCC [28]. Therefore, the two histological groups, CC and GR, were combined in our analysis and henceforth referred to as GRCC. Follow-up subgroup analysis of CC and GR cases were also conducted separately.

The following variables were collected and coded: AYA site recode, primary site, ICD-O-3 histology, age at diagnosis, sex, race, grade, ER status, PR status, HER2 status, immunohistochemistry (IHC) based intrinsic subtype of breast cancer, American Joint Commission on Cancer (AJCC) 6th Edition Staging, AJCC 6th Edition tumor size (T), lymph nodes affected (N), and metastasis (M) Staging, bone metastasis at diagnosis, brain metastasis at diagnosis, liver metastasis at diagnosis, lung metastasis at diagnosis, surgery at primary site, and radiation. Details of coding for hormone receptor and HER2 status are available on the SEER website (https://seer.cancer.gov/seerstat/databases/ssf/her2-derived.html) and in the AJCC Collaborative Stage Data Collection System Coding Manual [38]. Cases diagnosed at autopsy or that could have 0 days of follow-up were excluded.

### 2.3. Statistical Analysis

All statistical analyses was carried out using the IBM SPSS Statistics software package (version 25, International Business Machines Corporation, Armonk, NY, USA). Differences in demographic and clinical characteristics between GRCC and non-GRCC cases were determined using the Pearson’s chi-square test and column proportions were compared using Bonferroni correction to adjust for multiple comparisons. Median survival times were determined using the Kaplan-Meier method and the significance of the difference was determined using the log-rank test. Multivariable analyses of overall survival were conducted using the Cox Proportional Hazards Ratios (HR) model. Corresponding hazard ratios and 95% confidence intervals (CI) were estimated from the model. Two-tailed *p*-values <0.05 were considered statistically significant.

## 3. Results

### 3.1. Demographics and Clinical Characteristics

Our SEER Program query identified 1,256,901 cases of malignant breast carcinoma. From these cases of interest, cases diagnosed at autopsy or that could have 0 days of follow-up (*n* = 5162, 0.4%) were excluded. The final number of cases included in our analysis was 1,251,739. Out of these, 110 cases (0.008%) were identified as CC and 45 of the cases (0.003%) were classified as GR. According to the WHO, the two terms are synonymous when referring to breast carcinomas [28] and since each patient could only be classified by one ICD code, both CC and GR cases were combined and henceforth referred to as GRCC. We first described our findings in the GRCC combined population. We then conducted subgroup analyses on GR and CC carcinomas, separately comparing these cases to non-GRCC cases and to each other; those results will be described subsequently.

Among all carcinomas of the breast, the median follow-up time was 60 months (range: 0 to 395 months), with 414,019 recorded deaths. In the GRCC population, the median follow-up time was 54 months (range: 2 to 96 months), with 63 deaths. The demographical and clinical characteristics of the patient population are summarized in Table 1. The median age at diagnosis of GRCC of the breast was 62 years old compared to 60 years old in non-GRCC carcinomas of the breast (*p* = 0.46). The majority of patients with GRCC carcinoma were female (98.1%) and ethnically white (80.6%), and 92.9% of GRCC patients received surgery and 45.7% received radiation therapy. We found that grade (*p* < 0.001), ER status (*p* <0.001), PR status (*p* < 0.001), IHC based intrinsic subtypes (*p* < 0.001), AJCC 6th staging (*p* = 0.03), T status (*p* = 0.01) and brain metastasis (*p* = 0.03) significantly differed between GRCC and non-GRCC carcinomas of the breast; although only a single case of GRCC with brain metastasis was available in our analysis. GRCC were statistically more likely to be Grade III (GRCC: 41.3% vs. non-GRCC carcinomas: 29.2%) and grade IV (GRCC: 3.2% vs. non-GRCC carcinomas: 1.4%), ER negative (GRCC: 40.3% vs. non-GRCC carcinomas: 17.9%), PR negative (GRCC: 58.3% vs. non-GRCC carcinomas: 27.4%), triple negative (GRCC: 44.8% vs. non-GRCC carcinomas: 10.4%), T2 status (GRCC: 36.8% vs. non-GRCC carcinomas: 23.6%), and positive for brain metastasis at diagnosis (GRCC: 3.4% vs. non-GRCC carcinomas: 0.4%).

### 3.2. Survival

The median survival time for GRCC patients was 158 months vs. 176 months for non-GRCC breast carcinomas (*p* = 0.02, Figure 1). The corresponding 5-, 10- and 15-year survival rates for GRCC were 70%, 53%, and 44%, respectively, whereas the 5-, 10-, 15-year survival rate for non-GRCC carcinomas was 79%, 64%, and 51%. After adjusting for age, disease stage, tumor grade, ER status, PR status, HER2 status, surgery status, and radiation treatment, survival for GRCC remained significantly poorer compared to non-GRCC carcinomas (HR: 1.33; 95% CI: 1.04–1.67; *p* = 0.025).

Among GRCC patients, older age (*p* = 0.002), higher AJCC stage (*p* < 0.001), T status (*p* < 0.001), N status (*p* = 0.001), and M status (*p* < 0.001) were also associated with significantly poorer survival (Table 2, Figure 2A), whereas surgery (*p* < 0.001) and radiation treatments (*p* = 0.02) significantly improved survival (Figure 2B left two panels). We further assessed the combination of surgery and radiation treatment on patient survival (Figure 2B, right two panels). We identified a significant survival improvement in patients who underwent sub-mastectomy—defined as any surgery less than a total mastectomy, plus radiation treatment compared to individuals who only underwent sub-mastectomy (*p* = 0.02). However, when total mastectomy was combined with radiation treatment, no improvement in survival was observed (*p* = 0.89). We found that age (HR: 1.05; 95% CI: 1.02–1.07; *p* < 0.001); AJCC stage III (HR: 4.67; 95% CI: 1.84–11.87; *p* = 0.001); AJCC stage IV (HR: 15.84; 95% CI: 4.11–60.97; *p* < 0.001); and radiation (HR: 0.47; 95% CI: 0.25–0.90; *p* = 0.02) remained significant prognostic factors for survival in our subsequent multivariable analysis after accounting for age, AJCC stage, surgery, or radiation (Table 3).

### 3.3. Subgroup Analysis of GR and CC Carcinoma of the Breast

To confirm our finding in the combined GRCC cases, we also conducted subgroup analyses on the GR and CC cases, comparing them to non-GRCC separately. The 45 cases of GR carcinoma had a median age at diagnosis of 61 years. The demographical and clinical characteristics of this patient subpopulation showed similar trends to those previously described in GRCC carcinomas of the breast (Appendix A). We found that grade (*p* = 0.04), ER status (*p* = 0.02), PR status (*p* = 0.003), receptor subtype (*p* = 0.002), T status (*p* = 0.02) and brain metastasis (*p* < 0.001) remained significantly differed between GR and non-GRCC carcinomas of the breast. AJCC 6th staging had a borderline significance (*p* = 0.06). The median survival time for GR was 133 months, with 5-, 10- and 15-year survival rates of 75%, 56%, and 45%, respectively. However, these survival durations no longer differed significantly from non-GRCC patients (*p* = 0.50, Appendix A). When comparing survival based on various demographic/clinical factors for the GR subgroup, only N staging reached statistical significance (*p* < 0.02, Appendix A). AJCC stage (*p* = 0.12), T status (*p* = 0.06), surgery (*p* = 0.06), and radiation (*p* = 0.10), factors which were identified as prognostic factors in our analysis of GRCC carcinomas, showed a consistent trend towards an effect on survival in the GR subgroup (Appendix A). Radiation after sub-mastectomy, however, remained significant in improving survival in GR patients (*p* = 0.04, Appendix A).

The 110 CC cases also displayed similar trends in demographical and clinical characteristics as the GRCC combined group (Appendix A). The only exceptions were that CC patients were more likely to be male than non-GRCC carcinoma patients of the breast (2.7% vs. 0.7%, *p* =0.012) and that they were no longer significantly different from non-GRCC patients with regards to brain metastasis at diagnosis (*p* = 0.31). Among CC cases, 5-, 10, and 15- year survival rates were 75%, 56%, and 45%, respectively. Median survival duration was 158 months, which was significantly worse than non-GRCC carcinomas of the breast (176 months in other carcinomas, p = 0.02, Appendix A). CC patients displayed poorer survival compared to non-GRCC patients after accounting for age, stage, tumor grade, ER/PR/HER2 status, surgery and radiation (HR: 1.33; 95% CI: 1.00–1.76; *p* = 0.047). Prognostic factors for CC cases included age (*p* = 0.006), AJCC stage (*p* < 0.001), T status (*p* = 0.04), N status (*p* = 0.02), M status (*p* < 0.001), surgery (*p* < 0.001) and radiation (*p* = 0.05) (Appendix A). Age (*p* < 0.001) and stage (*p* < 0.001) remained independent prognostic factors of poorer survival in multivariable analysis (Appendix A).

When comparing CC directly to the GR subgroup, there were no significant differences in demographical and clinical characteristics (Appendix A) or survival durations (*p* = 0.59, Appendix A).

## 4. Discussion

Using the SEER Program database, our study identified that GRCC is an aggressive histology with a poorer prognosis than non-GRCC carcinomas of the breast. Most importantly, the poorer prognosis was irrespective of the AJCC stage, tumor grade, patient age, treatment, and molecular subtypes defined by ER/PR/HER2 status. Our findings of poorer prognosis in GRCC carcinomas of the breast mirrors the poorer outcomes of clear cell carcinomas of other origins, such as the kidney [39], uterus [40], and ovaries [41]. We further identified that prognostic factors for non-GRCC breast carcinomas, such as age and AJCC staging, including TNM staging, were also relevant for GRCC tumors. Moreover, treatment with radiation and surgery showed improvements in survival for GRCC cases. Specifically, radiation was associated with improved survival when less aggressive surgical procedures (i.e., breast-conserving surgeries) were performed. This finding is consistent with the standard of care for other breast carcinomas [42] and suggests that GRCC may be treated similarly. While similar trends were observed in the GR subgroup, statistical significance was not reached, likely due to the smaller number of cases as compared to the GRCC analysis. In the CC cases, the significance of two demographic/clinical factors differed from our GRCC analysis possibly due to variations in the selected cases.

Several biomolecular theories have been investigated as the reasoning for poorer prognosis of GRCC tumors. Aberrant glycogen can act as an additional fuel source in nutrient-deprived tumor microenvironments. Tumor cells have been shown to have the ability to mobilize glycogen to fuel glycolysis and cellular proliferation via a p38α mitogen-activated protein kinase (p38α-MAPK) signaling pathway [43]. Hypoxia-induced glycogen phosphorolysis (the breakdown of glycogen into glucose) has also been found to enhance tumorigenesis by suppressing reactive oxygen species levels and p53-dependent senescence [13]. Furthermore, glycogen accumulation in tumor-associated stroma was identified recently to play key roles in tumor microenvironment by supplying carbohydrate for tumor proliferation [43]. Although further investigations are needed, elevated levels of glycogen content in breast cancer may participate in similar signaling pathways to potentiate tumor growth.

Triple-negative breast tumors account for 15–20% of all breast cancers and their highly aggressive nature is well documented [44,45]. These tumors have poorer prognosis due to both the lack of receptors available for targeted therapies and they also have higher metabolic demands for cell division. Basu et al. demonstrated significantly higher fluorine-18 fluorodeoxyglucose (FDG) uptake in triple negative tumors compared with uptake in ER+/PR+/HER2− tumors using FDG-positron emission tomography [46]. Studies have also demonstrated a greater glycolytic phenotype and altered mitochondrial respiration in triple negative patient tumors and cell lines compared with other tumor subtypes [47,48,49]. These data are suggestive of a more metabolically active phenotype, likely to compensate for increased cellular proliferation rates. High glycogen content could be utilized to satisfy these demands when the solid tumor grows beyond the limit of nutrient diffusion. Although our multivariable analysis showed that the poorer prognosis of GRCC was irrespective of IHC intrinsic receptor subtype, the increased metabolic needs of triple negative tumors suggest that it may still be a factor in influencing the poorer prognosis of GRCC. However, only 29 cases were available for analysis as HER2 receptor status was only available after 2010 in the SEER database. Therefore the relationship between glycogen and triple negative tumors requires further examination.

The most widely accepted technique for glycogen histochemistry is PAS staining [50]. However, PAS not only detects glycogen; it also stains other polysaccharides such as glycoprotein, proteoglycans, and mucins [51,52,53]. Another disadvantage of PAS is the lack of sensitivity; only large polysaccharide inclusions can be identified by PAS. Given that our database analysis identified only a small number of GR (45 cases) compared to CC (110 cases), we speculate that glycogen accumulations may be under-identified due to the lack of PAS sensitivity. Routine glycogen staining may also be underutilized as clinicians do not feel the need to additionally assess the specific intracytoplasmic moiety as no specific treatment guidelines currently exist for GRCC. However, the poorer prognostic outcome for GRCC tumors illustrated in our study highlights the utility of accurate subtyping of invasive carcinomas in order to improve our understanding of the prognostic variables. Detailed histological characterization will also provide clinicians with an opportunity for detailed assessment of treatment outcomes depending on the subtype of the disease. Given the importance of glycogen in patient survival and cancer prognosis, more specific and accurate methods for glycogen detection may be warranted.

While less common than glycogen accumulations, various histological subtypes of CC carcinoma without glycogen can exist. Apocrine, lipid-rich, pleomorphic lobular (histiocytoid) or secretory carcinomas all can have a histologically “clear cell” appearance [27,32]. Consistent with previous studies, the lack of differentiation between the CC and GR cases in our subgroup analysis provides further evidence that the majority of CC cases are expected to be GR. As we were unable to confirm the presence of glycogen accumulation in the CC carcinomas of our retrospective database analysis, larger studies of CC carcinomas with validated GR cases are needed to confirm the clinical and prognostic factors identified in our study. Due to the rarity of GRCC cases, only a single positive case was available for our analysis of bone, brain, liver and lung metastasis at the time of diagnosis. Although GRCC reached a statistically higher chance of brain metastasis when compared to non-GRCC, future studies with a larger number of positive cases are needed to confirm these results. Furthermore, chemotherapy information was not available in the SEER program database. A follow-up study that includes chemotherapy treatment would further our understanding of the disease and establish better treatment options.

## 5. Conclusions

While it has previously been difficult to accurately assess clinical characteristics and prognosis of GRCC due to the rarity of their presentation, using the SEER program database we were able to examine 155 of these cases. Our results demonstrated that GRCC is an aggressive histology with a poorer prognosis irrespective of AJCC stage, tumor grade, patient age, treatment, and ER/PR/HER2 status. To further clarify the characteristics and prognosis of GRCC of the breast, a large, systematic study of histologically confirmed glycogen-rich cases with long-term follow up will be necessary. As it has not been possible to study these rare tumors in randomized clinical trials to determine optimal surgical, radiation or chemotherapeutic strategies, more sensitive histological characterization methods may be warranted in order to further identify prognostic variables and to promote the development of targeted therapies for these patients.

## Figures and Tables

**Figure 1 jcm-08-00246-f001:**
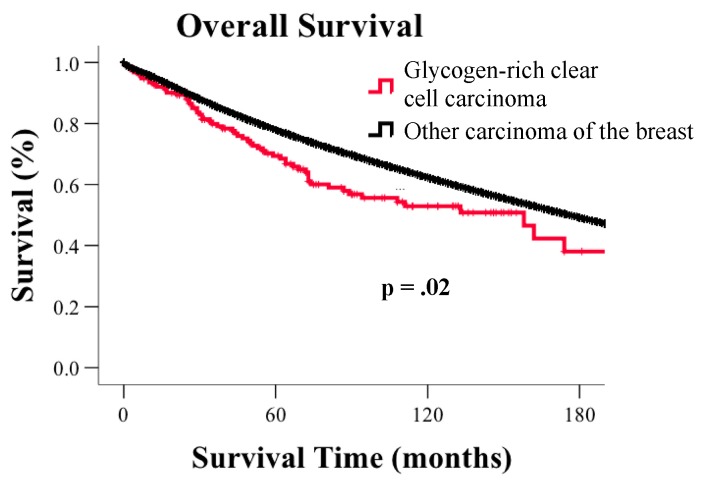
Kaplan-Meier curves for overall survival based on histological subtype.

**Figure 2 jcm-08-00246-f002:**
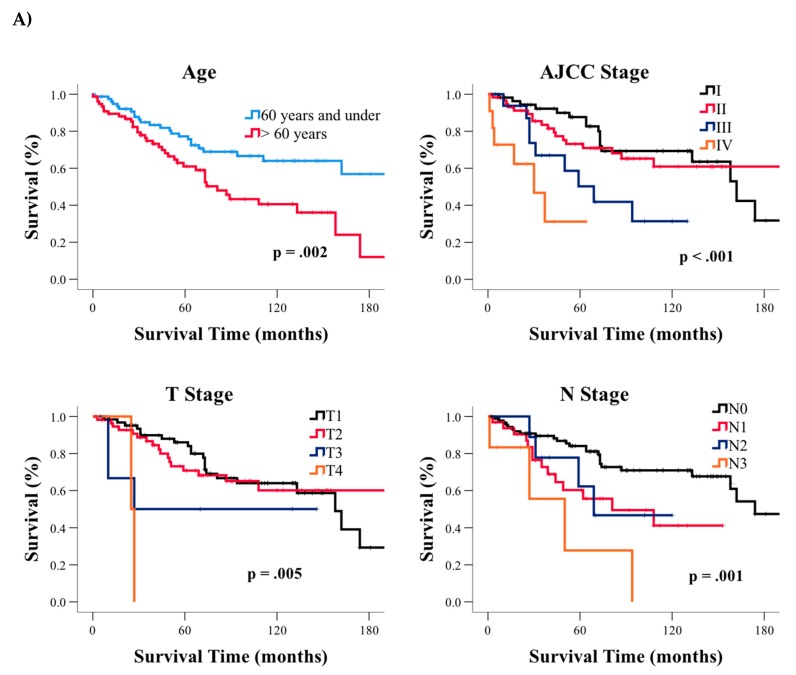
Kaplan-Meier curves for glycogen-rich clear cell (GRCC) based on (**A**) clinical factors and (**B**) treatment. AJCC—American Joint Committee on Cancer; T, N, M refers to tumor size, nodal status, and metastasis classifications of malignant tumor staging system, respectively; RT—radiation therapy.

**Table 1 jcm-08-00246-t001:** Demographical and clinical characteristics.

		GRCC		Non-GRCC		
		Count	%	Count	%	*p*-Value
Age	0–60 years	78	50.3%	592,544	47.3%	0.46
	>60 years	77	49.7%	659,040	52.7%	
Sex	Female	152	98.1%	1,242,647	99.3%	0.07
	Male	3	1.9%	8937	0.7%	
Race	American Indian/Alaska Native	1	0.6%	6099	0.5%	0.98
	Asian or Pacific Islander	12	7.7%	85,407	6.8%	
	Black	16	10.3%	123,081	9.8%	
	Unknown	1	0.6%	5967	0.5%	
	White	125	80.6%	1,031,030	82.4%	
Grade	Well differentiated; Grade I	5	3.2%	216,525	17.3%	**<0.001**
	Moderately differentiated; Grade II	53	34.2%	446,827	35.7%	
	Poorly differentiated; Grade III	64 *	41.3%	365,903	29.2%	
	Anaplastic; Grade IV	5 *	3.2%	16,937	1.4%	
	Unknown	28	18.1%	205,392	16.4%	
ER Status ^a^	Negative	58 *	40.3%	203,173	17.9%	**<0.001**
	Positive	67	46.5%	807,640 *	71.1%	
	Borderline	0	0.0%	2837	0.2%	
	Unknown	19	13.2%	121,600	10.7%	
PR Status ^a^	Negative	84 *	58.3%	310,973	27.4%	**<0.001**
	Positive	40	27.8%	683,610 *	60.2%	
	Borderline	0	0.0%	5794	0.5%	
	Unknown	20	13.9%	134,873	11.9%	
HER2 Status ^c^	Negative	25	86.2%	291,238	78.5%	0.56
	Positive	2	6.9%	52,021	14.0%	
	Borderline	0	0.0%	8250	2.2%	
	Unknown	2	6.9%	19,359	5.2%	
IHC based intrinsic Subtypes ^c^	HER2−/HR+	12	41.4%	251,946 *	67.9%	**<0.001**
	HER2+/HR−	1	3.4%	15,597	4.2%	
	HER2+/HR+	1	3.4%	36,269	9.8%	
	Triple Negative	13 *	44.8%	38,750	10.4%	
	Unknown	2	6.9%	28,306	7.6%	
AJCC 6th Stage	I	54	34.8%	525,652	42.0%	**0.03**
	II	59 *	38.1%	365,536	29.2%	
	III	17	11.0%	139,296	11.1%	
	IV	11	7.1%	54,270	4.3%	
	Unknown	14	9.0%	166,830	13.3%	
T	T0	0	0.0%	1110	0.1%	**0.01**
	T1	63	40.6%	647,840 *	51.8%	
	T2	57 *	36.8%	295,198	23.6%	
	T3	7	4.5%	49,233	3.9%	
	T4	3	1.9%	33,931	2.7%	
	Tis	0	0.0%	1791	0.1%	
	Unknown	25	16.1%	222,481	17.8%	
N	N0	91	58.7%	718,198	57.4%	0.79
	N1	32	20.6%	234,449	18.7%	
	N2	9	5.8%	68,563	5.5%	
	N3	6	3.9%	48,862	3.9%	
	Unknown	17	11.0%	181,512	14.5%	
M	M0	132	85.2%	1,083,182	86.5%	0.21
	M1	11	7.1%	54,270	4.3%	
	Unknown	12	7.7%	114,132	9.1%	
Bone Metastasis ^c^	No Metastasis	27	93.1%	351,795	94.9%	0.72
	Metastasis	1	3.4%	13,264	3.6%	
	Unknown	1	3.4%	5809	1.6%	
Brain Metastasis ^c^	No Metastasis	27	93.1%	363,212	97.9%	**0.03**
	Metastasis	1 *	3.4%	1501	0.4%	
	Unknown	1	3.4%	6155	1.7%	
Liver Metastasis ^c^	No Metastasis	27	93.1%	359,794	97.0%	0.46
	Metastasis	1	3.4%	5079	1.4%	
	Unknown	1	3.4%	5995	1.6%	
Lung Metastasis ^c^	No Metastasis	27	93.1%	358,514	96.7%	0.56
	Metastasis	1	3.4%	6,198	1.7%	
	Unknown	1	3.4%	6,156	1.7%	
Surgery ^b^	No Surgery	7	5.5%	70,372	7.3%	0.38
	Surgery	118	92.9%	889,255	92.0%	
	Unknown	2	1.6%	6822	0.7%	
Extent of Surgery ^b^	No Surgery	8	6.3%	75,877	7.9%	0.51
	Sub-Mastectomy ^#^	70	55.1%	485,610	50.2%	
	Mastectomy	48	37.8%	401,380	41.5%	
	Surgery, Unknown	0	0.0%	1358	0.1%	
	Unknown	1	0.8%	2224	0.2%	
Radiation ^b^	No Radiation	64	50.4%	426,196	44.1%	0.21
	Radiation	58	45.7%	513,445	53.1%	
	Unknown	5	3.9%	26,808	2.8%	

Bolded are statistically significant *p*-values when comparing between clear cell/glycogen rich to other carcinomas of the breast. * Statistically significant differences between column proportions. ^#^ Defined as any surgery less aggressive than total mastectomy. ^a,b,c^ Variable available for cases diagnosed after 1990, 1998 and 2010 respectively. GRCC—glycogen-rich clear cell; ER—estrogen receptor; PR—progesterone receptor; IHC—immunohistochemistry; HER2—human epidermal growth factor receptor 2; HER2+—human epidermal growth factor receptor 2 positive; HER2−—human epidermal growth factor receptor 2 negative; HR+—hormone receptor positive; HR−—hormone receptor negative; AJCC—American Joint Committee on Cancer; T, N, M refers to tumor size, nodal status, and metastasis classifications of malignant tumor staging system, respectively.

**Table 2 jcm-08-00246-t002:** Survival durations for glycogen-rich clear cell (GRCC) carcinoma patients.

		Median (Months)	95% CI		*p*-Value
			Lower	Upper	
All GRCC cases		158	96.5	219.5	**NA**
Age	60 years and under	158	96.6	219.4	**0.002**
	>60 year	73	18.6	127.4	
Sex	Female	-	-	-	0.56
	Male	81	62.5	99.5	
Race	American Indian/Alaska Native	27	-	-	0.16
	Asian or Pacific Islander	-	-	-	
	Black	-	-	-	
	White	111	45.6	176.4	
Grade	Well to Moderately Differentiated	162	105.1	218.9	0.19
	Poorly Differentiated to Anaplastic	111	-	-	
ER Status	Negative	158	69.3	246.7	0.82
	Positive	174	61.2	286.8	
PR Status	Negative	158	86.3	229.7	0.15
	Positive	-	-	-	
AJCC Stage	I	162	125.7	198.3	**<0.001**
	II	-	-	-	
	III	69	38.4	99.6	
	IV	30	7.9	52.1	
T	T1	158	125.4	190.6	**0.005**
	T2	-	-	-	
	T3	27	-	-	
	T4	25	-	-	
N	N0	174	-	-	**0.001**
	N1	81	15.3	146.7	
	N2	69	-	-	
	N3	50	12.3	87.7	
M	M0	162	121.1	202.9	**<0.001**
	M1	30	7.9	52.1	
Surgery	No Surgery	25	0.0	58.1	**<0.001**
	Surgery	158	100.4	215.6	
Extent of Surgery	No Surgery	25	0.0	58.1	**0.02**
	Sub-Mastectomy ^#^	174	116.0	232.0	
	Mastectomy	108	-	-	
Radiation	No Radiation	81	40.4	121.6	**0.012**
	Radiation	-	-	-	

HER2 Status was only available for cases diagnosed after 2010 and was therefore excluded from the analysis. Bolded are statistically significant *p*-values. “-” Median survival time not reached or could not be calculated. ^#^ Defined as any surgery less aggressive than total mastectomy. CI—confidence interval; ER—estrogen receptor; PR—progesterone receptor; AJCC—American Joint Committee on Cancer; NA—not applicable; T, N, M refers to tumor size, nodal status, and metastasis classifications of malignant tumor staging system, respectively.

**Table 3 jcm-08-00246-t003:** Multivariable analysis of overall survival for glycogen-rich clear cell (GRCC.) patients.

Variable	Reference	*p*-Value	HR	95% CI	
				Lower	Upper
Age		**<0.001**	1.05	1.02	1.07
AJCC Stage	I	**<0.001**			
	II	0.64	1.18	0.60	2.34
	III	**0.001**	4.67	1.84	11.87
	IV	**<0.001**	15.84	4.11	60.97
Surgery	No Surgery	0.39	2.05	0.40	10.45
Radiation	No Radiation	**0.02**	0.47	0.25	0.90

Bolded are statistically significant *p*-values. CI—confidence interval; HR—hazards ratio; AJCC—American Joint Committee on Cancer.

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
