# Peer review of "Clinical Features, Survival and Prognostic Factors of Glycogen-Rich Clear Cell Carcinoma (GRCC) of the Breast in the U.S. Population"

_jcm, 2019, doi:10.3390/jcm8020246_

Round 1
Reviewer 1 Report
The study is the first comprehensive study of its kind and therefore will be valuable for furthering knowledge in GRCC. Data has been presented clearly although figure 2 could be redesigned for clarity. The figure is really hard to read and is very pixelated when magnified. Otherwise the study is an useful addition to existing literature in GRCC.
Author Response
The authors would like to thank the reviewer for his/her favorable comments. We are delighted that the reviewer also shares our enthusiasm for the novelty and clinical implications of our work and agree with the reviewer that the article will be a great asset to the GRCC field.
We acknowledge the poor resolution for Figure 2. We have redesigned the figure using enlarged versions of the Kaplan Meier curves with higher resolution.
Reviewer 2 Report
In this manuscript Zhou and colleagues analyze the clinical features of glycogen-rich clear cell breast carcinomas. This data is important because little is known about this disease, and a large analysis of its features have not yet been done. The data is well presented, however some things should be addressed:
1. The data provided here should be discussed more in the discussion section. Half of the discussion describes possibilities of why you did not have a lot of GRCC samples.
2. The authors state that the WHO describes CC and GR as synonymous with each other and then analyze the diseases together for most of the paper. Please comment on why the authors then analyze the diseases separately later in the paper if they would be treated the same in the clinic.
3. I understand that the rareness of this disease limits the number of samples available for analysis, but some of the conclusions of significance are based upon 1 sample in the GRCC being positive for said characteristic. Therefore, something should be mentioned about this in the paper.
4. Some of the p-values in the survival plots are very difficult to read.
Author Response
We thank the Reviewer for his/her thoughtful comments. Please see below for detailed, point-by-point responses to address each of Reviewer’s concerns. Do not hesitate to let us know if there are any additional questions/concerns.
1. The data provided here should be discussed more in the discussion section. Half of the discussion describes possibilities of why you did not have a lot of GRCC samples.
The most striking finding in our manuscript is the poorer prognosis with GRCCs. Therefore, we have added an additional paragraph in the discussion section describing metabolic changes found in triple negative breast cancers. In conjunction with paragraph 2 of discussion, we examine multiple possible causes of the more aggressive phenotype with glycogen accumulation.
2. The authors state that the WHO describes CC and GR as synonymous with each other and then analyze the diseases together for most of the paper. Please comment on why the authors then analyze the diseases separately later in the paper if they would be treated the same in the clinic.
This was done as a control to confirm CC and GR were equivalent to each other with regards to clinical features, survival and prognostic factors. Although WHO describes CC and GR as synonymous in these factors, we understand that the two categories are based on different histological definitions, therefore, we wanted to further confirm whether they were indeed equivalent in the cases we analyzed. The lack of significant differences between CC and GR groups detected in our subgroup analysis was consistent with WHO’s definitions and our expectations.
3. I understand that the rareness of this disease limits the number of samples available for analysis, but some of the conclusions of significance are based upon 1 sample in the GRCC being positive for said characteristic. Therefore, something should be mentioned about this in the paper.
We thank the Reviewer for bringing up this valuable point. We have modified our results section to highlight when only a single case of a particular characteristic was available for GRCC i.e. bone, brain, liver and lung metastases. We have also modified the last paragraph of our discussion section to include it as an additional limitation of our results. Additionally, we have removed the increased likelihood of brain metastasis from the conclusions described in our abstract.
4. Some of the p-values in the survival plots are very difficult to read.
We have redesigned figure 2 using an enlarged version of the Kaplan Meier curves with higher resolution.